# Multi-Bellman operator for convergence of $Q$-learning with linear function approximation

**Diogo S. Carvalho**                                    *diogo.s.carvalho@tecnico.ulisboa.pt*
*INESC-ID*
*Instituto Superior Técnico, University of Lisbon*

**Pedro A. Santos**                                       *pedro.santos@tecnico.ulisboa.pt*
*INESC-ID*
*Instituto Superior Técnico, University of Lisbon*

**Francisco S. Melo**                                         *fmelo@inesc-id.pt*
*INESC-ID*
*Instituto Superior Técnico, University of Lisbon*

**Reviewed on OpenReview:** *https://openreview.net/forum?id=D2PjEPGXgh*

## Abstract

We investigate the convergence of $Q$-learning with linear function approximation and introduce the multi-Bellman operator, an extension of the traditional Bellman operator. By analyzing the properties of this operator, we identify conditions under which the projected multi-Bellman operator becomes a contraction, yielding stronger fixed-point guarantees compared to the original Bellman operator. Building on these insights, we propose the multi-$Q$-learning algorithm, which achieves convergence and approximates the optimal solution with arbitrary precision. This contrasts with traditional $Q$-learning, which lacks such convergence guarantees. Finally, we empirically validate our theoretical results.

## 1 Introduction

Reinforcement learning aims to approximate the value of performing different actions in different states, considering the expected sum of time-discounted rewards in a Markovian environment. The importance of this task cannot be overstated, as an accurate value function allows an agent to make optimal decisions by selecting the action with the highest value in a given state (Puterman, 2005). The value function also facilitates environment evaluation and enables comparisons between different environments. If we can store the value of performing each action on each state individually, the $Q$-learning algorithm converges to the correct value function (Watkins & Dayan, 1992). When it is not possible to store values, for example when there are too many states, $Q$-learning can be combined with function approximation (Melo & Ribeiro, 2007).

Unfortunately, the combination of $Q$-learning and function approximation is troublesome, and even when the function approximation space is linear, the algorithm can diverge (Sutton & Barto, 2018). In fact, the approximation problem that $Q$-learning addresses does not have, in general, a solution (Melo et al., 2008). To address this limitation, we propose an alternative problem and algorithm that effectively solve linear function approximation in reinforcement learning, unlike $Q$-learning. The problem and algorithm proposed are extensions of the original. To our ends, we start by introducing the multi-Bellman operator, and identify the conditions under which its projection into the function approximation space is contractive. Leveraging the contractivity property of the Multi-Bellman operator, we propose the multi $Q$-learning algorithm, and demonstrate its convergence to arbitrarily good solutions, both theoretically and empirically.

## 2 Background

A Markov decision problem (MDP) is a tuple $(\mathcal{X}, \mathcal{A}, \mathcal{P}, r, \gamma)$, where $\mathcal{X}$ is a discrete set of states, $\mathcal{A}$ is a finite set of actions, $\mathcal{P}$ is a set of distributions over $\mathcal{X}$ for each state and action, $r : \mathcal{X} \times \mathcal{A} \to [0, 1]$ is a reward function, and $\gamma$ is a real in $[0, 1)$ called the discount factor (Puterman, 2005). A policy is a function $\pi : \mathcal{X} \to \Delta(A)$, mapping states to distributions over $\mathcal{A}$, and its value is the function $q_\pi : \mathcal{X} \times \mathcal{A} \to \mathbb{R}$ such that

$$q_\pi(x, a) = \mathbb{E}\left[\sum_{t=0}^{\infty} \gamma^t r_t \mid x_0 = x, a_0 = a\right], \tag{1}$$

where the expectation is with respect to trajectories of the MDP, namely rewards $r_t$ and states $x_{t+1}$ that are obtained by performing action $a_t$ on state $x_t$, with actions $a_t$ that are selected according to $\pi$ at every $t$. There is at least one policy $\pi^*$ that maximizes the value $q$ on every state and action (Puterman, 2005) and we refer to its value as $q^*$. The value $q^*$ satisfies the Bellman fixed-point equation given by

$$q^* = \mathrm{H}q^*, \tag{2}$$

where H is the Bellman operator, such that

$$(\mathrm{H}q)(x, a) = \mathbb{E}\left[r(x, a) + \gamma \max_{a' \in \mathcal{A}} q(x', a')\right], \tag{3}$$

where the expectation is with respect to the next state $x'$ obtained by performing action $a$ on state $x$.

### 2.1 Function Approximation

Our goal is to approximate $q^*$. In other words, we want to find a good parameterized representation of the value of an optimal policy for a given MDP, given a function approximation space $\mathcal{H} = \{h_\omega : \mathcal{Z} \to \mathbb{R}, \omega \in \mathbb{R}^k\}$. Let us consider that every $h_\omega \in \mathcal{H}$ is differentiable in $\omega$, a function $h : \mathcal{Z} \to \mathbb{R}$, a distribution $\mu$ over a random $z$ taking values in $\mathcal{Z}$, and the loss $l(\omega) = \frac{1}{2}\|h - h_\omega\|^2$, where we consider the $\mu$-norm such that $\|h\|^2 = \mathbb{E}_\mu\left[h^2(z)\right]$. To approximate $h$ is to find an element in the set $\mathrm{Proj}\, h \subset \mathcal{H}$ such that

$$\mathrm{Proj}\, h = \underset{h_\omega \in \mathcal{H}}{\arg\min}\, l(\omega). \tag{4}$$

Any $\omega$ parameterizing a $h_\omega$ in $\mathrm{Proj}\, h$ must be a critical point and thus verify

$$\nabla_\omega l(\omega) = 0. \tag{5}$$

**Linear function approximation**  Given features $\phi : \mathcal{Z} \to \mathbb{R}^k$, a linear function approximation space is given by the functions such that $h_\omega(z) = \phi^T(z)\omega$. In this case, the gradient of the loss function is given by

$$\nabla_\omega l(\omega) = -\mathbb{E}_\mu\left[\phi(z)\left(h(z) - h_\omega(z)\right)\right]. \tag{6}$$

Solving for $\nabla_\omega l(\omega) = 0$, we obtain that

$$\omega = \mathbb{E}_\mu\left[\phi(z)\phi^T(z)\right]^{-1} \mathbb{E}_\mu\left[\phi(z)h(z)\right]. \tag{7}$$

Thus, if the inverted matrix above exists, the set $\mathrm{Proj}\, h$ has a single element and we can refer to both as $h_{\omega^*}$. The solution $\omega^*$ is also the equilibrium of the dynamical system

$$\dot{\omega} = \nabla_\omega l(\omega), \tag{8}$$

and the limit of the sequence of $\omega_t$ obtained by performing a discretized update with $z_t$ i.i.d. from $\mu$

$$\omega_{t+1} = \omega_t + \alpha_t\left[\phi(z_t)\left(h(z_t) - h_{\omega_t}(z_t)\right)\right], \tag{9}$$

with $\alpha_t$ a small positive real called the learning rate.

**Stochastic approximation**   Without access to $h$, stochastic approximation performs the update

$$\omega_{t+1} = \omega_t + \alpha_t \left[ \phi\left(z_t\right)\left(\tau_t - h_{\omega_t}(z_t)\right) \right], \tag{10}$$

with $\tau_t$ a possibly $\omega_t$-dependent estimate of $h\left(z_t\right)$ called the target. If there exists an equilibrium $\omega^*$ for the dynamical system

$$\dot{\omega} = \mathbb{E}_\mu \left[ \phi\left(z_t\right)\left(\tau_t - h_\omega(z_t)\right) \right], \tag{11}$$

and it is globally asymptotically stable, well-established conditions guarantee the stochastic approximation update converges to $\omega^*$ (Borkar, 2008).

## 2.2   $Q$-learning with Linear Function Approximation

We focus now on the reinforcement learning setting and consider $\mathcal{Z} = \mathcal{X} \times \mathcal{A}$, and an approximation space of linearly parameterized functions $q_\omega : \mathcal{X} \times \mathcal{A} \to \mathbb{R}$ such that

$$q_\omega(x, a) = \phi^T(x, a)\omega. \tag{12}$$

In reinforcement learning, we want to approximate the value $q^*$, by computing $\omega^*$ that satisfies

$$q_{\omega^*} = \operatorname{Proj} q^*. \tag{13}$$

Without knowledge of $q^*$, we are unable to perform the exact stochastic approximation update,

$$\omega_{t+1} = \omega_t + \alpha_t \left[ \phi\left(x_t, a_t\right)\left(q^*(x_t, a_t) - q_{\omega_t}(x_t, a_t)\right) \right]. \tag{14}$$

Leveraging the fact that $q^* = \mathrm{H}q^*$, $Q$-learning performs instead the stochastic approximation update

$$\omega_{t+1} = \omega_t + \alpha_t \left[ \phi\left(x_t, a_t\right)\left(\tau_t - q_{\omega_t}(x_t, a_t)\right) \right], \tag{15}$$

where the target $\tau_t$ is a sample for $\left(\mathrm{H}q_{\omega_t}\right)\left(x_t, a_t\right)$, i.e.

$$\tau_t = r_t + \gamma \max_{a' \in \mathcal{A}} q_{\omega_t}\left(x_t', a'\right). \tag{16}$$

Since the target is not differentiated, despite possibly depending on $\omega_t$, Sutton & Barto (2018) calls these semi-gradient methods. An equilibrium for $Q$-learning with linear function approximation must be the fixed-point for the projected Bellman operator, thus verifying the fixed-point equation

$$q_\omega = \operatorname{Proj}\left(\mathrm{H}q_\omega\right). \tag{17}$$

Unfortunately, since the Projected Bellman operator is not generally a contraction, the fixed-point may not exist and, even when it does, the solution may not be an equilibrium of the associated dynamical system

$$\dot{\omega} = \mathbb{E}_\mu \left[ \phi\left(x_t, a_t\right)\left(\tau_t - q_\omega(x_t, a_t)\right) \right]. \tag{18}$$

Consequently, $Q$-learning with linear function approximation can diverge, failing to provide any stable solution. The divergence of $Q$-learning is evidenced in classic counter-examples where the parameters of the approximator do not approach any solution, either oscillating within a window (Boyan & Moore, 1995; Gordon, 2001) or growing without bound (Tsitsiklis & Van Roy, 1996; Baird, 1995). Currently, the theoretical results that establish convergence of $Q$-learning restrict the data or the features too much (Szepesvári & Smart, 2004; Melo et al., 2008), and the proposed variants of $Q$-learning that are guaranteed to converge under more general conditions (Carvalho et al., 2020; Zhang et al., 2021; Lim et al., 2022) converge to fundamentally biased limit solutions that do not hold good performance guarantees (Chen et al., 2023).

## 3 Multi-Bellman Operator

Let us define the multi-Bellman operator $\mathrm{H}^n$, such that

$$\mathrm{H}^1 q = \mathrm{H}q \tag{19}$$

and

$$\mathrm{H}^{n+1}q = \mathrm{H}\left(\mathrm{H}^n q\right). \tag{20}$$

Since $q^* = \mathrm{H}q^*$, it also holds that, for every $n$ in $\mathbb{N}$,

$$q^* = \mathrm{H}^n q^*. \tag{21}$$

**Proposition 3.1.** *The operator $\mathrm{H}^n$ is a contraction in the $\infty$-norm with contraction factor $\gamma^n$.*

*Proof.* See the proof of B.1 of the appendix. $\square$

While unsurprising, the result has profound implications on the use of linear function approximation.

**Assumption 3.2.** The features and data are such that the matrix $\mathbb{E}_\mu\left[\phi(x,a)\phi^T(x,a)\right]$ is invertible.

**Theorem 3.3.** *There exists $N \in \mathbb{N}$ such that, for all $n \geq N$, the operator $\mathrm{Proj}\,\mathrm{H}^n$ is a contraction in the $\infty$-norm.*

*Proof.* See the proof of B.2 of the appendix. $\square$

**Corollary 3.4.** *There exists $N \in \mathbb{N}$ such that, for all $n \geq N$, there exists a unique solution $\tilde{\omega}^n$ to the fixed-point equation*

$$q_\omega = \mathrm{Proj}(\mathrm{H}^n q_\omega). \tag{22}$$

The result states that, while the projected Bellman operator may not be contractive, the projected multi-Bellman operator is guaranteed to be contractive for sufficiently large $n$. Consequently, whereas the fixed-point equation of the projected Bellman operator may fail to have a solution, the fixed-point equation of the projected multi-Bellman operator is guaranteed to have a unique solution.

While the previous result establishes existence and uniqueness of solution, it says nothing about the quality of such solution in light of the reinforcement learning objective. Now, we compare $q_{\tilde{\omega}^n}$ with the optimal solution.

**Assumption 3.5.** For all $t \in \mathbb{N}$, $(x_t, a_t)$ is i.i.d. from $\mu$.

**Theorem 3.6.** *For all $n \geq N$, the solution $\tilde{\omega}^n$ is such that*

$$\|q^* - q_{\tilde{\omega}^n}\|_\infty \leq \frac{1}{1 - \lambda\gamma^n} \|q^* - q_{\omega^*}\|_\infty. \tag{23}$$

*where $\lambda = \sigma_{\max}\phi_{\max}^2$, with $\phi_{\max} = \max_{x,a}\|\phi(x,a)\|_2$ and $\sigma_{\max} = \left\|\mathbb{E}_\mu\left[\phi(x,a)\phi^T(x,a)\right]^{-1}\right\|_2$.*

*Proof.* See the proof of B.3 of the appendix. $\square$

**Corollary 3.7.** *The sequence of $\{\tilde{\omega}\}_{n \geq N}$ is such that*

$$\lim_{n \to \infty} \|\omega^* - \tilde{\omega}^n\| = 0. \tag{24}$$

The results state that the solution $q_{\tilde{\omega}}$ can be made arbitrarily close to the optimal solution $q_{\omega^*}$, and can therefore provide an optimal approximation of $q^*$ in a given function approximation space. In the following, we design a stochastic approximation algorithm that aims to approximate $\tilde{\omega}^n$.

## 4    Multi $Q$-learning

For given $n \in \mathbb{N}$, to approximate the solution of

$$q_\omega = \text{Proj}(\text{H}^n q_\omega), \tag{25}$$

we propose to perform the multi $Q$-learning update

$$\omega_{t+1} = \omega_t + \alpha_t \left[ \phi(x_t, a_t) \left( \tau_t^{n,m} - q_{\omega_t}(x_t, a_t) \right) \right], \tag{26}$$

where the target $\tau_t^{n,m} := \tau^{n,m}(x_t, a_t)$ is a sample for $(\text{H}^n q_{\omega_t})(x_t, a_t)$ defined recursively as:

$$\tau^{1,m}(x_t, a_t) = r_t + \gamma \max_{a'} q_{\omega_t}(x_t', a') \tag{27}$$

and

$$\tau^{n+1,m}(x_t, a_t) = r_t + \gamma \max_{a'} \left( \frac{1}{m} \sum_{i=1}^{m} \tau_i^{n,m}(x_t', a') \right), \tag{28}$$

where $\tau_i^{n,m}(x_t', a')$ are i.i.d. $\tau^{n,m}(x_t', a')$ and are used to build an average with $m$ samples.

At time step $t$, instead of building a 1-step greedy target, multi $Q$-learning builds a target that is obtained by trying, on every state encountered along $n$-step trajectory starting at $x_t'$, every action $m$ times, and maximizing the average of the sum of the discounted rewards at each level. We assume access to a fixed replay buffer $\mathcal{D}$, from which state-action pairs are sampled, and a simulator to sample transitions. Our algorithm is most similar to the one of Kearns et al. (2002). In our case, in addition to off-policy learning, we have function approximation and bootstrapping. Our algorithm is also similar to the one of Rosenberg et al. (2023), but with off-policy learning and function approximation. More generally, our $n$-step updates differ from the usual in that they are policy-independent, which as we will see allows our updates to approximate the multi-Bellman operator accurately. Moreover, our setting introduces additional technical challenges due to maximization at each step and at each level, which does not usually appear in the related works and can lead to biased updates and solution if not properly handled. We provide a pseudo-code of Multi $Q$-learning in Algorithm 1, which calls a function that builds the target for the update defined in Algorithm 2.

---

**Algorithm 1** Multi $Q$-learning

---

**Initialize** weight vector $\omega_0$
**Given** a fixed replay buffer $\mathcal{D}$ containing state-action pairs $(s_t, a_t)$
**Access to a simulator** that returns $(r_t, s_t')$ given $(s_t, a_t)$
**For each training iteration:**

- **Sample** a random state-action pair $(s_t, a_t)$ from $\mathcal{D}$

- **Simulate** reward $r_t$ and next state $s_t'$ by calling the simulator with $(s_t, a_t)$

- **Call** build_target to compute the target:

$$\tau^{n,m}(s_t, a_t) = \text{build\_target}(n, m, s_t, a_t, \omega_t)$$

- Update weights:
$$\omega_{t+1} \leftarrow \omega_t + \alpha \left[ \phi(s_t, a_t) \left( \tau^{n,m}(s_t, a_t) - q_{\omega_t}(s_t, a_t) \right) \right]$$

---

### 4.1    Convergence

Contrary to the case of $Q$-learning, there is a solution to the dynamical system of the multi $Q$-learning update.

---

**Algorithm 2** build_target function

---

**Input:** parameters $n$, $m$, state $s$, action $a$, weight vector $\omega$
**Output:** Target $\tau$
**Simulate** reward $r_{s,a}$ and next state $s'$ by calling the simulator with $(s, a)$
**If** $n = 1$:

- **Compute** the target:
$$\tau = r_{s,a} + \gamma \max_{a' \in \mathcal{A}} q_\omega(s', a')$$

**Else:**

- **For each** $i = 1 \ldots m$:
    - **Recursively obtain sample:** $\tau_i^{n-1,m}(s', a') = \text{build\_target}(n - 1, m, s', a', \omega)$
- **Compute** the target:
$$\tau = r_{s,a} + \gamma \max_{a' \in \mathcal{A}} \sum_{i=1}^{m} \tau_i^{n-1,m}(s', a')$$

**Return** $\tau$

---

**Proposition 4.1.** *There exists $N \in \mathbb{N}$ such that, for all $n \geq N$, there exists an equilibrium $\hat{\omega}^{n,m}$ for the dynamical system*

$$\dot{w} = \mathbb{E}_\mu \left[\phi(x, a) \left(\tau^{n,m}(x, a) - q_\omega(x, a)\right)\right] \tag{29}$$

*Proof.* See the proof of C.1 of the appendix. □

We show that multi $Q$-learning converges to the solution.

**Assumption 4.2.** *The sequence of learning rates $\{\alpha_t\}_{t \in \mathbb{N}}$ satisfies $\sum_{t=0}^{\infty} \alpha_t = \infty$ and $\sum_{t=0}^{\infty} \alpha_t^2 < \infty$.*

**Theorem 4.3.** *There exists $N \in \mathbb{N}$ such that, for all $n \geq N$ and $m \in \mathbb{N}$, the sequence of $\{\omega_t\}_{t \in \mathbb{N}}$ is such that*

$$\lim_{t \to \infty} \|\hat{\omega}^{n,m} - \omega_t\| = 0. \tag{30}$$

*Proof.* See the proof of C.2 of the appendix. □

The result establishes conditions under which multi $Q$-learning converges to $q_{\hat{\omega}^{n,m}}$. The assumptions are commonplace (Tsitsiklis & Van Roy, 1996; Carvalho et al., 2020; Melo et al., 2008). Assumptions 3.2 and 4.2 are mild. Assumption 3.5 is the most restrictive of the three, as it requires the data distribution to have no shift during training. Same as the other authors, we use the assumption to facilitate our technical analysis of the convergence of $Q$-learning, which still fails in this benign setting. Nevertheless, the replay buffer can slow down the data distribution shift arbitrarily much. In the limit, the case of offline reinforcement learning, the distribution is indeed fixed and samples are i.i.d.. We do believe our result can be extended to the case where the data distribution converges.

### 4.2 Performance

Unfortunately, in general, $q_{\hat{\omega}^{n,m}}$ does not equal $q_{\tilde{\omega}^n}$. Specifically, when the environment is not deterministic, the target of multi $Q$-learning is biased, due to the difficulty in estimating the maximum of an expectation. In fact, it is impossible to estimate the expectation of the maximum without bias (Van Hasselt, 2013). We show, however, that as the number $m$ of samples per action grows, regardless of the randomness of the environment, the solution of multi $Q$-learning converges to the unbiased solution $q_{\tilde{\omega}^n}$.

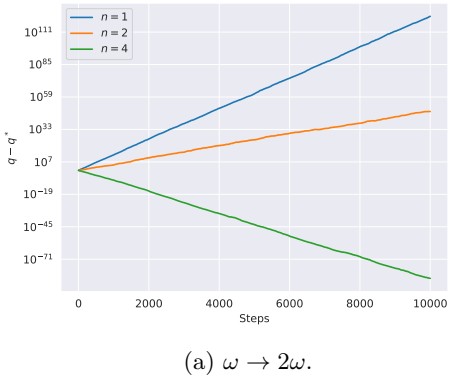
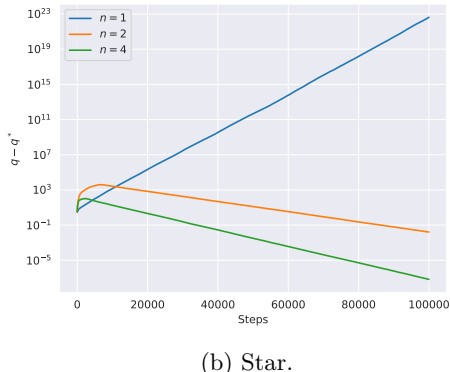

(a) $\omega \to 2\omega$.    (b) Star.

Figure 1: Classic counter-examples. The $y$-axis shows the difference between the estimated $q$ and $q^*$. In both examples, $Q$-learning, that is multi $Q$-learning with $n = 1$, diverges but, for sufficiently large $n$, multi $Q$-learning converges.

**Theorem 4.4.** *For all $n \geq N$ and $m \in \mathbb{N}$, we have that*

$$\left\| q_{\tilde{\omega}^n} - q_{\hat{\omega}^{n,m}} \right\| \leq \frac{\xi}{1 - \lambda \gamma^n} \frac{\max \sqrt{\mathrm{var}\left(\tau^{n-1,m}\right)}}{\sqrt{m}}, \tag{31}$$

*with $\xi = \lambda \frac{\sqrt{|\mathcal{A}| - 1}}{1 - \gamma}$.*

*Proof.* See the proof of C.8 of the appendix. □

The results states that, for each $n$, when $m$ increases, the solution of multi $Q$-learning $q_{\hat{\omega}^{n,m}}$ approaches the fixed-point of the projected Multi-Bellman operator $q_{\tilde{\omega}^n}$. On the other hand, it states that, for each $m$, when $n$ increases, the distance between the two can grow, but is bounded, since we also have that $\sqrt{\mathrm{var}\left(\tau^{n-1,m}\right)} \leq \frac{1}{1-\gamma}$ for all $n$ and $m$.

**Corollary 4.5.** *The sequence of $\{\hat{\omega}^{n,m}\}_{m \in \mathbb{N}}$ is such that*

$$\lim_{m \to \infty} \left\| \tilde{\omega}^n - \hat{\omega}^{n,m} \right\| = 0. \tag{32}$$

From Corollary 3.7, we also knew that, as $n$ grows, the fixed-point of the projected Multi-Bellman operator $q_{\tilde{\omega}^n}$ approaches the optimal solution $q_{\omega^*}$. Putting the two results together, we conclude that, increasing $n$ and $m$, the solution of multi $Q$-learning becomes arbitrarily close to $q_{\omega^*}$.

## 5 Experiments

In this section, we validate our theoretical findings. We start by focusing on the convergence property of multi $Q$-learning. All results are averages of five runs with standard deviations. Given the linear function approximation setting and relatively small scale of the environments, all experiments can be performed on standard commercial CPUs, with small memory costs, and lasting less than eight hours.

### 5.1 Convergence

We show that, on well-known problems where $Q$-learning with linear function approximation fails to converge, multi $Q$-learning converges if $n$ is sufficiently large, as predicted in Theorem 4.3. We start by using two classical problems that are typically used in the literature to exhibit the divergence of $Q$-learning with linear function approximation. The first problem was introduced by Tsitsiklis & Van Roy (1996) and the second

problem by Baird (1995). Even though both problems are relatively simple and even though the correct solution $q^*$ is within the function approximation space, the parameters of $Q$-learning diverge to infinity in some conditions. Additionally, we introduce another simple example, which we call the Bias example. This example serves as a complement to the first examples by showcasing the bias that can appear in multi $Q$-learning, as predicted in 4.4, and how we can minimize it.

$\omega \to 2\omega$  The $\omega \to 2\omega$ is a problem due to Tsitsiklis & Van Roy (1996). The MDP has two states $y_1$ and $y_2$ and only one action $b_1$. Performing the action on any of the two states always takes the agent to the second state. The reward received is always zero. Therefore, $q^*$ is zero. The features $\phi : \mathcal{X} \times \mathcal{A} \to \mathbb{R}$ such that $\phi(y_1, b_1) = 1$ and $\phi(y_2, b_1) = 2$. We consider a discount factor of 0.9, which is within the interval where $Q$-learning is originally reported to diverge, and a learning rate of $10^{-1}$, and uniform data distribution. Figure 2 depicts the problem.

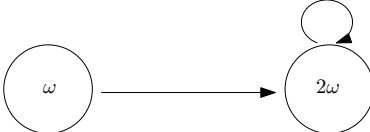

Figure 2: Transition diagram of the process in the $\omega \to 2\omega$ problem.

**Star**  The Star problem is due to Baird (1995). The MDP has seven states $y_1$ to $y_7$ and two actions $b_1$ and $b_2$. The first action always takes the agent to the last state, the second action takes the agent to any of the first five states uniformly. The reward received is always zero. Therefore, $q^*$ is zero. The features are $\phi : \mathcal{X} \times \mathcal{A} \to \mathbb{R}^{16}$ such that, for $j$ between 1 and 7, for all $i$ between 1 and 16 $\phi_i(y_j, b_2) = \mathbf{1}(i = j + 1)$, for $i$ between 2 and 7 $\phi_i(y_j, b_1) = 2 \cdot \mathbf{1}(i = j + 1)$, for $j$ between 1 and 6 $\phi_1(y_j, b_1) = \mathbf{1}(j \le 6)$ and $\phi_1(y_7, b_1) = 2$, $\phi_i(y_j, b_1) = 0$ otherwise. We consider a discount factor of 0.98, within the interval at which $Q$-learning diverges, a learning rate of $10^{-2}$, and a data distribution that samples the first action one seventh of the times and the second action six sevenths of the times. Figure 3 depicts the problem.

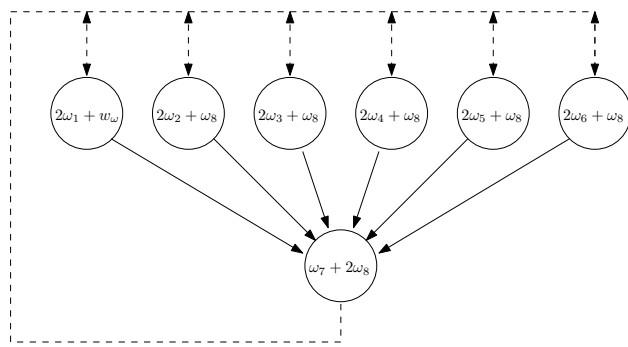

Figure 3: Transition diagram of the process in the Star problem.

We test multi $Q$-learning for $n \in \{1, 2, 4\}$. We note that $n = 1$ recovers the original $Q$-learning. We use $m = 1$. The experimental results in Figure 1 validate our theoretical finding that, whereas the projected Bellman operator may fail to have a fixed-point and $Q$-learning may diverge, multi $Q$-learning converges for large enough $n$, as predicted by Theorem 4.3, to the correct solution $q^* = 0$.

**Bias**  In some non-deterministic environments that are not reward-free, the solution of multi-Q-learning may not equal the projected multi-Bellman operator, resulting in a bias that increases with $n$. To highlight this behavior, we introduce the Bias example. The Bias example is intentionally simple, consisting of only two states and two actions with uniformly random transitions. One state provides a reward of 1, while the other provides 0. This example serves two key purposes. First, it serves to illustrate the potential bias in

multi $Q$-learning when $n > 1$. As shown in prior work (Van Hasselt, 2013), it is generally impossible to estimate the maximum of a set of expected values of random variables without introducing bias. In our case, for fixed $n$ and $m$, multi $Q$-learning can converge to a biased solution of the equation $q = \text{Proj}(\text{H}^n q)$, and this bias increases with $n$. This phenomenon aligns with the predictions of our theoretical results (Theorem 4.4). Second, the Bias problem serves to demonstrate how increasing the parameter $m$ effectively reduces the bias. Our theoretical results (Corollary 4.5) suggest that increasing $m$—the number of samples used to construct the update target—reduces this bias. The example provides a controlled setting to validate this claim.

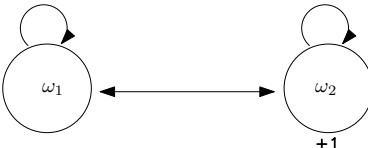

Figure 4: Transition diagram of the process in the Bias problem.

We use tabular features, with a discount factor of 0.9 and a learning rate of 0.1. Figure 4 shows the transition diagram for the Bias problem. The results in Figure 5a confirm that, for $m = 1$, the bias increases as $n$ increases. In contrast, Figure 5b shows that, as $m$ increases, the bias decreases, aligning with the theoretical predictions in Corollary 4.5.

## 5.2 Performance

We also validate that, as $n$ increases, the distance between the fixed-point of the projected Bellman operator and $q_{\omega^*}$ decreases, as predicted by Corollary 3.7. We use an $\epsilon$-greedy policy where $\epsilon$ decays linearly from 100% to 5% during the first half of interactions and remains constant afterwards. We use a replay buffer with 20% of the total number of timesteps used for the environment. We plot a moving average of the last 5% of the total number of time steps. We use classic deterministic environments, which we describe.

**Acrobot** Acrobot is a classic control problem proposed by Sutton (1995) where a joint actuates two links such that one end is fixed and the other is free. The actions of the agent are to apply a negative torque to the joint, apply a positive torque to the joint or do nothing. The state space is the tuple composed of sine, cosine and angular velocity of each link. The reward is always minus one unless the free end of the links reaches a target height, after which the agent reaches a terminal state. We discretize each dimension of the state space in four and use gaussian features. We use a discount factor of 0.99 and a learning rate of $3 \cdot 10^{-3}$.

**Cartpole** Cartpole is a classic control problem proposed by Barto et al. (1983), where a cart balances a pole. The actions of the agent are to push the cart left or right. The state space is a tuple with the position and velocity of the cart and the angle and angular velocity of the pole. The reward is always one unless the pole falls and the agent reaches a terminal state. We discretize each dimension of the state space in two and use gaussian features. We use a discount factor of 0.99 and a learning rate of $3 \cdot 10^{-2}$.

The results in Figure 6 validate the finding that as $n$ increases, the solution of multi $Q$-learning improves, as predicted by Corollary 3.7.

Altogether, our experimental results confirm that, contrary to the original $Q$-learning, multi $Q$-learning converges for sufficiently large $n$, that the distance between the projected multi-Bellman operator and the optimal $\text{Proj}\, q^*$ decreases with $n$, and that the distance between the solution of multi $Q$-learning and the fixed-point of the projected multi-Bellman operator decreases with $m$.

## 6 Related Work

We analyze work related to our own, both with respect to the problem of $Q$-learning with linear function approximation and our proposed multi $Q$-learning solution.

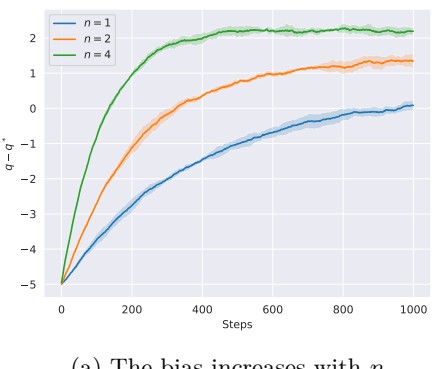

(a) The bias increases with $n$.

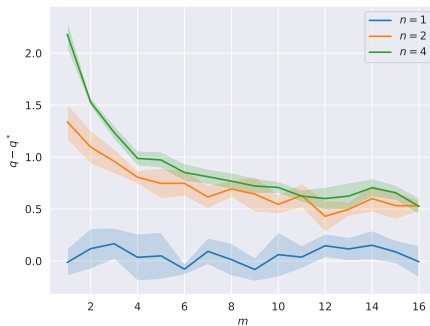

(b) The bias decreases with $m$.

Figure 5: Bias problem.

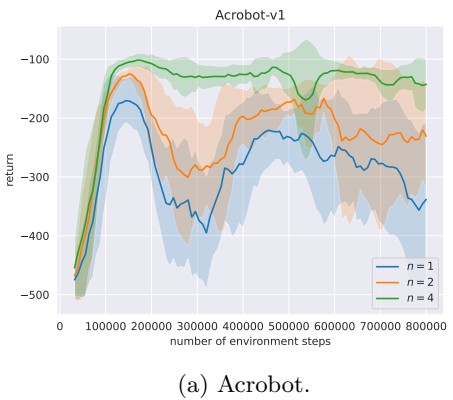

(a) Acrobot.

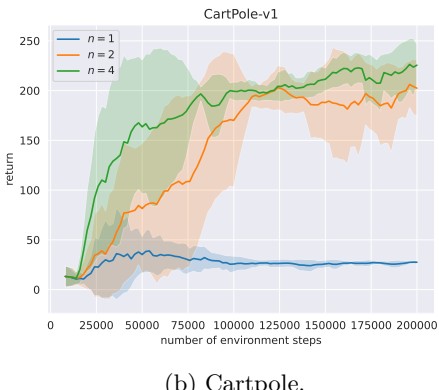

(b) Cartpole.

Figure 6: Classic control. The $y$-axis shows the average return. The performance increases with $n$.

**Convergence conditions**   Around the time the divergence of $Q$-learning was established, some works aimed at reaching conditions under which $Q$-learning would converge. The first works proposed restrictions of the function approximation spaces themselves. They establish that with some specific choices of features, $Q$-learning is guaranteed to converge (Singh et al., 1994; Ormoneit & Sen, 2002; Szepesvári & Smart, 2004). The linear architectures considered across the three works are extensions of what would be the one-hot representation in the tabular case. Afterwards, another work considered again general linear function approximation architectures. Melo et al. (2008) prove that $Q$-learning with linear function approximation converges if the distribution of state-action pairs the agent uses to learn is sufficiently close to the distribution that the optimal policy induces.

**Gradient-TD methods**   Instead of finding conditions under which $Q$-learning converges, a different line of works proposes to take a step back and modify the objective that $Q$-learning is trying to solve. Maei et al. (2010) propose to perform full-gradient descent on a different objective, called the projected Bellman error. The resulting algorithm, called Greedy-GQ, is a gradient-TD methods. Being a full gradient method, Greedy-GQ is provably convergent to a minimum in the approximation space. However, the method can converge to local minima and there is no guarantee that the resulting greedy policy is a good control policy (Scherrer, 2010). Gradient-TD is also less efficient than semi-gradient methods (Mahadevan et al., 2014; Du et al., 2017).

**Regularized methods**   The problem of divergence of $Q$-learning with function approximation was significantly revived after an empirical success story of $Q$-learning with deep neural networks (Mnih et al., 2015).

One of the components of the renowned deep $Q$-network (DQN) therein is a target network that aims at compensating the instability generated by the $Q$-learning updates. While the DQN is not provably convergent, its empirical success inspired theoretical results. The works of Carvalho et al. (2020), Zhang et al. (2021) and Chen et al. (2023) provided convergence guarantees for variants with target networks. Additionally, another work points out that the target network can be seen as a regularizer (Piché et al., 2021). As hinted by Farahmand (2011), several works prove that various forms of regularization of the $Q$-values or the parameters themselves can stabilize $Q$-learning, resulting in a convergent algorithms (Zhang et al., 2021; Lim et al., 2022; Agarwal et al., 2021). However, the introduction of the regularizers biases the solution obtained, even in deterministic environments. Contrarily, the bias of our method only appears in non-deterministic environments and can be minimized arbitrarily much.

**Non-linear function approximation** The behaviour of $Q$-learning with linear function approximation has been the focus of several theoretical works. In practice, however, $Q$-learning is mostly used with non-linear function approximation, especially through neural networks (Mnih et al., 2015). Still, there are works that address this more general setting. A recent work suggests a loss function that is decreasing over time, assuming the neural network converges to a target network at each step (Wang & Ueda, 2021). However, having a loss function that is monotonically decreasing does not imply that neither the parameters of the approximator are converging nor that the $Q$-values are converging. Cai et al. (2019) and Xu & Gu (2020) provide finite-time analysis of $Q$-learning with over-parameterized neural networks that convergence happens only as the size of the network grows to infinity, which is an impractical scenario.

**Lookahead** In the context of policy iteration, several works hinted at the theoretical benefits of lookahead in the tabular case (De Asis et al., 2018; Efroni et al., 2018a). Efroni et al. (2018b) then identified a problem with soft policy improvement for lookahead policies, which happens if function approximation is used. Specifically, contrarily to what happens with a single-step policy improvement step, a multistep policy improvement step is not necessarily monotonically increasing. However, Efroni et al. (2020) and Winnicki & Srikant (2022) respectively provide finite-time and asymptotic results for lookahead in approximate policy iteration. Rosenberg et al. (2023) proposes an interesting policy-based algorithm with adaptive depth that results in convergence in the tabular case. We focus on the problem of divergence of $Q$-learning, a value-based algorithm, when used with linear function approximation architectures; the projected multi-Bellman operator that we introduced differs in that it is designed for evaluating the optimal policy. Chen et al. (2022) also analyze an operator similar to our proposed multi-Bellman operator in the context of off-policy prediction in an actor-critic setting with linear function approximation, leveraging the improved fixed-point guarantees. Due to the maximization that appears on the Bellman operator for control, in our case we lose the property of linearity and, since a maximization and an expectation do not, in general, commute, the $Q$-learning setting is more challenging and the algorithm we propose is significantly different.

**Planning and model-based reinforcement learning** Multi-step real time dynamic programming algorithms, as defined by Efroni et al. (2018b) and discussed by Moerland et al. (2020), integrate planning and learning and have several successful practical applications (Silver et al., 2017; 2018). We refer to the survey of Moerland et al. (2023) for varied algorithms that include $Q(\sigma)$ (De Asis et al., 2018), tree-backup (Precup, 2000) and multi-step expected SARSA (Sutton & Barto, 2018). While most such algorithms are policy-based and on-policy, with separate value and policy and behavior-dependent solutions, we have an off-policy, value-based algorithm, which plans exclusively at training time multi $Q$-learning. Our fixed depth and full breadth setting are under-explored (Moerland et al., 2023).

# 7 Conclusion

Our work has made significant contributions to addressing the convergence challenge in $Q$-learning with linear function approximation. By introducing the multi-Bellman operator and demonstrating its contractive nature, we have paved the way for improved convergence properties. The proposed multi $Q$-learning algorithm effectively approximates the fixed-point solution of the projected multi-Bellman operator. Importantly, we have shown that the algorithm converges under relatively mild conditions.

### 7.1 Limitations and Future Work

Even though our algorithm can be combined with models that are learned concurrently to the reinforcement learning update to simulate the environment, in our experiments, we used a given simulator of the environment. It is not the case that such simulator is always available in applications. There are several techniques for learning a model of the environment. For example, we can learn the model previously or concurrently to reinforcement learning; the model can be exact or inexact; the model can be parametric or non-parametric. Multi $Q$-learning can be combined with any of the mentioned model-based approaches to remove the need for a simulator when it is not available.

Multi $Q$-learning also performs every action on every reached state while building the target. Thus, the computational complexity is exponential on the depth $n$. In case only some actions are performed on each state, we cannot show convergence to an optimal solution. Nevertheless, in some cases, the computational benefits of not performing every action on every state may outweigh the theoretical comfort. An analysis of this trade-off would be valuable. Future work should also analyze techniques to prune the set of actions sampled at each level of multi $Q$-learning and reduce the complexity of the algorithm.

It would also be interesting to analyze multi $Q$-learning with non-linear function approximation, both theoretically and empirically. Specifically, under which conditions and function approximation architectures would our convergence result still hold? How would multi $Q$-learning with non-linear function approximation compare, in practice, with relevant policy-free reinforcement learning algorithms such as the DQN (Mnih et al., 2015)? And will the target of multi-$Q$-learning, if combined with the target network, increase the performance of the DQN? Finally, it would be interesting to see our method applied to real-world problems where the environment, namely the transitions and rewards, can be simulated, including the training of large language models (LLMs).

### Acknowledgments

This work was partially supported by the Portuguese Fundação para a Ciência e a Tecnologia (FCT) under INESC-ID multi-annual funding (UIDB/50021/2020), RELEvaNT (PTDC/CCICOM/5060/2021), WS-MART ROUTE+ (2022.04180.PTDC) and CRAI (C628696807-00454142 (IAPMEI/PRR)). Diogo S. Carvalho acknowledges his FCT PhD grant (2020.05360.BD).

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

## A  Stochastic approximation

**Theorem A.1.** *Let us suppose that the following hold for the stochastic approximation setting of Sec. 2.*

1. *The map $g : \mathbb{R}^k \to \mathbb{R}^k$ such that*

$$g(\omega) = \mathbb{E}\left[\phi\left(z\right)\left(\tau(\omega) - g_\omega(z)\right)\right]$$

   *is Lipschitz;*

2. *The sequence of $m_t$ such that*

$$m_{t+1} = \phi(z_t)\left(\tau_{t+1} - g_{\omega_t}(z_t)\right) - g(\omega_t);$$

   *is a martingale difference sequence with respect to $\{(\tau_s, \omega_s) : s \le t\}$ and is such that*

$$\mathbb{E}\left[\|m_{t+1}\|^2 \mid \{(\tau_s, \omega_s) : s \le t\}\right] \le c_m\left(1 + \|\omega_t\|^2\right).$$

3. *The o.d.e*

$$\dot{\omega} = g(\omega)$$

   *has a unique and globally asymptotically stable equilibrium $\omega^*$ such that*

$$\omega^* = \mathbb{E}\left[\phi\left(z\right)\phi^T\left(z\right)\right]^{-1}\mathbb{E}\left[\phi\left(z\right)\tau(\omega)\right];$$

4. *The map $g_c : \mathbb{R}^k \to \mathbb{R}^k$ such that*

$$g_c(\omega) = \frac{g(c\omega)}{c}$$

   *is such that $\lim_{c \to \infty}\|g_c(\omega) - g_\infty(\omega)\|$ uniformly on compacts for a $g_\infty : \mathbb{R}^k \to \mathbb{R}^k$ and that the origin is the unique and globally asymptotically stable equilibrium of*

$$\dot{\omega} = g_\infty(\omega).$$

*Then, the sequence of stochastic approximation updates $\omega_t$ converges to $\omega^*$.*

## B  Multi-Bellman operator

**Proposition B.1.** *The operator $\mathrm{H}^n$ is a contraction in the $\infty$-norm with contraction factor $\gamma^n$.*

*Proof.* We establish the result by induction. First, we show that H contracts in the $\infty$-norm. We start by writing the expression $\|\mathrm{H}q - \mathrm{H}p\|_\infty$ in simpler terms, where $q$ and $p$ are value functions.

$$\|\mathrm{H}q - \mathrm{H}p\|_\infty = \max_{x,a}\left|\mathbb{E}\left[r(x,a) + \gamma \max_{a' \in \mathcal{A}} q\left(x', a'\right)\right] - \mathbb{E}\left[r(x,a) + \gamma \max_{a' \in \mathcal{A}} p\left(x', a'\right)\right]\right|$$

$$= \gamma \max_{x,a}\left|\mathbb{E}\left[\max_{a' \in \mathcal{A}} q\left(x', a'\right) - \max_{a' \in \mathcal{A}} p\left(x', a'\right)\right]\right|.$$

Next, we make use of Jensen's inequality to establish that

$$\|\mathrm{H}q - \mathrm{H}p\|_\infty \leq \gamma \max_{x,a} \mathbb{E}\left[\left|\max_{a'\in\mathcal{A}} q\left(x',a'\right) - \max_{a'\in\mathcal{A}} p\left(x',a'\right)\right|\right].$$

We use that the absolute difference of maxima is less than or equal to the maxima of absolute difference.

$$\|\mathrm{H}q - \mathrm{H}p\|_\infty \leq \gamma \max_{x,a} \mathbb{E}\left[\max_{a'\in\mathcal{A}} |q\left(x',a'\right) - p\left(x',a'\right)|\right].$$

Then, we use that the expectation of a random function is less than or equal to its maximum.

$$\|\mathrm{H}q - \mathrm{H}p\|_\infty \leq \gamma \max_{x'\in\mathcal{X}} \max_{a'\in\mathcal{A}} |q\left(x',a'\right) - p\left(x',a'\right)|$$
$$= \gamma \max_{x',a'} |q\left(x',a'\right) - p\left(x',a'\right)|$$
$$= \gamma \|q - p\|_\infty$$

Now, we make the induction step for $\mathrm{H}^{n+1} = \mathrm{H}\mathrm{H}^n$, observing that

$$\|\mathrm{H}\left(\mathrm{H}^n q\right) - \mathrm{H}\left(\mathrm{H}^n p\right)\|_\infty \leq \gamma \|\mathrm{H}^n q - \mathrm{H}^n p\|_\infty$$
$$= \gamma\gamma^n \|q - p\|_\infty$$
$$= \gamma^{n+1} \|q - p\|_\infty.$$

We conclude the proof. $\qquad\square$

**Theorem B.2.** *There exists $N \in \mathbb{N}$ such that, for all $n \geq N$, $\mathrm{Proj}\,\mathrm{H}^n$ is a contraction in the $\infty$-norm.*

*Proof.* We want to show that there exists a function $\lambda : \mathbb{N} \to \mathbb{R}$ such that

$$\|\mathrm{Proj}\left(\mathrm{H}^n q\right) - \mathrm{Proj}\left(\mathrm{H}^n p\right)\|_\infty \leq \lambda(n) \|q - p\|_\infty$$

and such that, for all $n$ greater than an existing $N$, $\lambda(n)$ is strictly smaller than 1.

We start by showing that $\mathrm{Proj}$ satisfies

$$\|\mathrm{Proj}\,q - \mathrm{Proj}\,p\|_\infty \leq \phi_{\max}^2 \sigma_{\max} \|q - p\|_\infty.$$

To see that, we start by recalling that

$$\left(\mathrm{Proj}\,q\right)(x,a) = \phi^T(x,a)\mathbb{E}\left[\phi(x,a)\phi^T(x,a)\right]^{-1}\mathbb{E}\left[\phi(x,a)q(x,a)\right].$$

Then, we have that

$$\|\mathrm{Proj}\,q - \mathrm{Proj}\,p\|_\infty = \max_{x,a} |\phi^T(x,a)\mathbb{E}\left[\phi(x,a)\phi^T(x,a)\right]^{-1}\mathbb{E}\left[\phi(x,a)\left(q(x,a) - p(x,a)\right)\right]|.$$

Now, we make use of a Cauchy-Schwarz inequality to say that

$$\|\mathrm{Proj}\,q - \mathrm{Proj}\,p\|_\infty \leq \max_{x,a} \left\|\phi^T(x,a)\mathbb{E}\left[\phi(x,a)\phi^T(x,a)\right]^{-1}\right\|_2 \left\|\mathbb{E}\left[\phi(x,a)\left(q(x,a) - p(x,a)\right)\right]\right\|_2$$

and we can write, using the definition of the matrix norm induced by the Euclidean norm in $\mathbb{R}^k$

$$\|\mathrm{Proj}\,q - \mathrm{Proj}\,p\|_\infty \leq \max_{x,a} \left\|\phi^T(x,a)\right\| \left\|\mathbb{E}\left[\phi(x,a)\phi^T(x,a)\right]^{-1}\right\|_2 \cdot$$
$$\cdot \left\|\mathbb{E}\left[\phi(x,a)\left(q(x,a) - p(x,a)\right)\right]\right\|_2$$

We obtain that

$$\|\mathrm{Proj}\,q - \mathrm{Proj}\,p\|_\infty \leq \phi_{\max}\sigma_{\max} \cdot \mathbb{E}\left[\|\phi(x,a)\|_2 |q(x,a) - p(x,a)|\right]$$
$$\leq \phi_{\max}^2 \sigma_{\max} \|q - p\|_\infty.$$

To conclude the proof, we start by combining the inequality we just established and Proposition B.1.

$$\left\|\text{Proj}\left(\text{H}^n q\right) - \text{Proj}\left(\text{H}^n p\right)\right\|_\infty \leq \phi_{\max}^2 \sigma_{\max} \left\|\text{H}^n q - \text{H}^n p\right\|_\infty$$
$$\leq \phi_{\max}^2 \sigma_{\max} \gamma^n \left\|q - p\right\|_\infty$$

We finish the proof by using $\lambda(n) = \phi_{\max}^2 \sigma_{\max} \gamma^n$. It is true that $\lambda(n) < 1$ for all $n \geq N$ when we consider $N = \lceil -\log_\gamma \left(\phi_{\max}^2 \sigma_{\max}\right)\rceil$.[1] $\qquad \square$

**Theorem B.3.** *For all $n \geq N$, where $N$ is identified in Theorem 3.3, $\tilde{\omega}^n$ is such that*

$$\left\|q^* - q_{\tilde{\omega}^n}\right\|_\infty \leq \frac{1}{1 - \lambda\gamma^n} \left\|q^* - q_{\omega^*}\right\|_\infty, \tag{33}$$

*Proof.* Let us use the triangle inequality with vertex $q_{\omega^*}$.

$$\left\|q^* - q_{\tilde{\omega}^n}\right\|_\infty \leq \left\|q^* - q_{\omega^*}\right\|_\infty + \left\|q_{\omega^*} - q_{\tilde{\omega}^n}\right\|_\infty.$$

Now we focus on the second term of the right-hand side above. We have that

$$\left\|q_{\omega^*} - q_{\tilde{\omega}^n}\right\|_\infty = \left\|\text{Proj}\left(\text{H}^n q^*\right) - \text{Proj}\left(\text{H}^n q_{\tilde{\omega}^n}\right)\right\|_\infty.$$

From Proposition 3.3, we have that

$$\left\|q_{\omega^*} - q_{\tilde{\omega}^n}\right\|_\infty \leq \sigma_{\max}\phi_{\max}^2 \gamma^n \left\|q^* - q_{\tilde{\omega}^n}\right\|_\infty.$$

We can conclude the result taking algebraic operations, using $\lambda = \sigma_{\max}\phi_{\max}^2$. $\qquad \square$

## C  Multi $Q$-learning

**Theorem C.1.** *There exists $N \in \mathbb{N}$ such that, for all $n \geq N$, there exists an equilibrium $\hat{w}^{n,m}$ for the dynamical system*

$$\dot{\omega} = \mathbb{E}\left[\phi(x, a)\left(\tau^{n,m}(\omega) - q_\omega(x, a)\right)\right]$$

*Proof.* We want to solve

$$\dot{\omega} = 0$$

or, equivalently,

$$\mathbb{E}\left[\phi(x, a)\left(\tau^{n,m}(\omega) - q_\omega(x, a)\right)\right] = 0.$$

Taking linear operations, we have equivalently

$$\omega = \mathbb{E}\left[\phi(x, a)\phi^T(x, a)\right]^{-1}\mathbb{E}\left[\phi(x, a)\left(\tau^{n,m}(\omega)\right)\right]$$

We show existence and uniqueness of solution to the fixed-point equation by showing that the right-hand side is contractive. For starters, we have that

$$\left\|\mathbb{E}\left[\phi(x, a)\phi^T(x, a)\right]^{-1}\right\|_2 = \sigma_{\max}.$$

Then, for any $\delta \in \mathbb{R}^k$, using Jensen's and Cauchy-Schwarz inequality, we have that

$$\left\|\mathbb{E}\left[\phi(x, a)\phi^T(x, a)\right]^{-1}\mathbb{E}\left[\phi(x, a)\left(\tau^{n,m}(\omega) - \tau^{n,m}(\omega + \delta)\right)\right]\right\|_2$$
$$\leq \sigma_{\max}\mathbb{E}\left[\left\|\phi(x, a)\right\|_2 \left|\mathbb{E}\left[\tau^{n,m}(\omega) - \tau^{n,m}(\omega + \delta)\right]\right|\right]$$
$$\leq \sigma_{\max}\phi_{\max}\gamma^n \left\|q_\omega - q_{\omega+\delta}\right\|_\infty$$
$$\leq \sigma_{\max}\phi_{\max}^2\gamma^n \left\|\delta\right\|_2$$

Thus, if $n$ is large enough, the operator is contractive and the fixed-point equation has a unique solution. $\qquad \square$

---

[1] $\lceil \cdot \rceil : \mathbb{R}^+ \to \mathbb{N}$ is the ceiling function, giving the smallest natural that is larger or equal to its argument.

### C.1 Convergence

**Theorem C.2.** *There exists $N \in \mathbb{N}$ such that, for all $n \geq N$ and $m \in \mathbb{N}$, the sequence of $\{\omega_t\}_{t \in \mathbb{N}}$ is such that*

$$\lim_{t \to \infty} \|\hat{\omega}^{n,m} - \omega_t\|_2 = 0.$$

*Proof.* To establish the result, we verify that we are in the conditions where Theorem A.1, which gives conditions for the convergence of stochastic approximation algorithms, holds. We do so by proving the following lemmas, one for each of the assumptions of the theorem. □

**Lemma C.3.** *The map $g : \mathbb{R}^k \to \mathbb{R}^k$ such that*

$$g(\omega) = \mathbb{E}\left[\phi(x,a)\left(\tau^{n,m}(\omega) - q_\omega(x,a)\right)\right]$$

*is Lipschitz.*

*Proof.* We start by writing

$$\|g(\omega) - g(\theta)\|_2 = \|\mathbb{E}\left[\phi(x,a)\left(\tau^{n,m}(\omega) - q_\omega(x,a)\right)\right] - \mathbb{E}\left[\phi(x,a)\left(\tau^{n,m}(\theta) - q_\theta(x,a)\right)\right]\|_2$$
$$\leq \|\mathbb{E}\left[\phi(x,a)\left(\tau^{n,m}(\omega) - \tau^{n,m}(\theta)\right)\right]\|_2 + \|\mathbb{E}\left[\phi(x,a)\left(q_\theta(x,a) - q_\omega(x,a)\right)\right]\|_2.$$

We can take care of the second term by making use of Jensen's and Cauchy-Schwarz' inequality.

$$\|\mathbb{E}\left[\phi(x,a)\left(q_\theta(x,a) - q_\omega(x,a)\right)\right]\|_2 \leq \mathbb{E}\left[\|\phi(x,a)\|_2 |q_\theta(x,a) - q_\omega(x,a)|\right]$$
$$\leq \phi_{\max}\|q_\theta - q_\omega\|_\infty$$
$$\leq \phi_{\max}^2 \|\omega - \theta\|_2.$$

Finally, by considering the sum of the coefficients from the first (obtained in the proof of C.1) and second terms, we have

$$\|g(\omega) - g(\theta)\|_2 \leq \phi_{\max}^2 (1 + \gamma^n) \|\omega - \theta\|_2,$$

or that $g$ is Lipschitz continuous. □

**Lemma C.4.** *The sequence of $m_t$ such that*

$$m_{t+1} = \phi(x_t, a_t)\left(\tau_t^{n,m} - g_{\omega_t}(x_t, a_t)\right) - g(\omega_t)$$

*is a martingale difference sequence with respect to $\{(\tau_s^{n,m}, \omega_s) : s \leq t\}$ and is such that*

$$\mathbb{E}\left[\|m_{t+1}\|^2 \mid \{(\tau_s^{n,m}, \omega_s) : s \leq t\}\right] \leq c_m \left(1 + \|\omega_t\|^2\right).$$

*Proof.* A martingale difference sequence has zero expectation conditioned on the past. Such condition is evident once we consider Assumption 3.5. Then, a martingale difference sequence also has finite first moment. That becomes evident when we observe that every term on the definition is bounded. Finally, since, again, every term on the definition is also bounded, the second moment is bounded. □

**Lemma C.5.** *The o.d.e*

$$\dot{\omega} = g(\omega)$$

*has a unique and globally asymptotically stable equilibrium $\tilde{\omega}$ such that*

$$\tilde{\omega} = \mathbb{E}\left[\phi(x,a)\phi^T(x,a)\right]^{-1} \mathbb{E}\left[\phi(x,a)\tau^{n,m}(\tilde{\omega})\right].$$

*Proof.* We already know from Theorem C.1 that the equilibrium $\tilde{\omega}$ exists and is unique. Let us consider a Lyapunov function $l : \mathbb{R}^k \to \mathbb{R}$ such that $l(\omega) = \frac{1}{2} \|\tilde{\omega} - \omega\|^2$. Existence of $\tilde{\omega}$ is guaranteed from Banach's fixed-point theorem and Proposition 3.3. It is clear that $l(\omega) = 0$ if $\omega = \tilde{\omega}$ and that $l(\omega) > 0$ if $\omega \neq \tilde{\omega}$. Now, we show that $\dot{l}(\omega) < 0$ if $\omega \neq \tilde{\omega}$.

We start by noticing that $\dot{l}(\omega) = \nabla l(\omega) \cdot \dot{\omega}$. Since $\nabla l(\omega) = -(\tilde{\omega} - \omega)$ and $\dot{\omega} = g(\omega)$, we have that

$$\dot{l}(\omega) = -(\tilde{\omega} - \omega) \, g(\omega).$$

Since $g(\tilde{\omega}) = 0$, it is also true that

$$\dot{l}(\omega) = -(\tilde{\omega} - \omega) \, (g\,(\omega) - g\,(\tilde{\omega}))$$

For now, let us focus on the second term of the right hand side and write

$$g(\omega) - g(\tilde{\omega}) = \mathbb{E}\left[\phi(x,a) \left(\tau^{n,m}(\omega) - \tau^{n,m}(\tilde{\omega})\right)\right] - \mathbb{E}\left[\phi(x,a) \left(q_{\tilde{\omega}}(x,a) - q_\omega(x,a)\right)\right]$$

Let us make two assertions, one for each of the terms on the right-hand side of the equation. First,

$$\left\|\mathbb{E}\left[\phi(x,a) \left(\tau^{n,m}(\tilde{\omega}) - \tau^{n,m}(\omega)\right)\right]\right\|_2 = \left\|\mathbb{E}\left[\phi(x,a) \left(\tau^{n,m}(\tilde{\omega}) - \tau^{n,m}(\omega)\right)\right]\right\|_2.$$

Using Jensen's and Cauchy-Schwarz' inequality just as in the previous proof, we get that

$$\left\|\mathbb{E}\left[\phi(x,a) \left(\tau^{n,m}(\omega) - \tau^{n,m}(\tilde{\omega})\right)\right]\right\|_2 \leq \phi_{\max}^2 \gamma^n \|\tilde{\omega} - \omega\|$$

Second, we have that

$$\mathbb{E}\left[\phi(x,a) \left(q_{\tilde{\omega}}(x,a) - q_\omega(x,a)\right)\right] = \mathbb{E}\left[\phi(x,a)\phi^T(x,a)\right] (\tilde{\omega} - \omega).$$

Having established the two assertions mentioned, we make use of them in the following.

$$\begin{aligned}
\dot{l}(\omega) &= -(\tilde{\omega} - \omega)\mathbb{E}\left[\phi(x,a) \left(\tau^{n,m}(\omega) - \tau^{n,m}(\tilde{\omega})\right)\right] - \\
&\quad - (\tilde{\omega} - \omega)\mathbb{E}\left[\phi(x,a)\phi^T(x,a)\right] (\tilde{\omega} - \omega) \\
&\leq \left\|(\tilde{\omega} - \omega)\,\mathbb{E}\left[\phi(x,a) \left(\tau^{n,m}(\omega) - \tau^{n,m}(\tilde{\omega})\right)\right]\right\|_2 - \\
&\quad - (\tilde{\omega} - \omega)\mathbb{E}\left[\phi(x,a)\phi^T(x,a)\right] (\tilde{\omega} - \omega) \\
&\leq \phi_{\max}^2 \gamma^n \|\tilde{\omega} - \omega\|_2^2 - \lambda_{\min} \|\tilde{\omega} - \omega\|_2^2 \\
&= \left(\phi_{\max}^2 \gamma^n - \lambda_{\min}\right) \|\tilde{\omega} - \omega\|_2^2,
\end{aligned}$$

where $\lambda_{\min}$ is the smallest eigenvalue of the auto-correlation matrix $\mathbb{E}\left[\phi(x,a)\phi^T(x,a)\right]$. The expression is negative if n is large enough. That concludes the proof. $\qquad\square$

**Lemma C.6.** *The map $g_c : \mathbb{R}^k \to \mathbb{R}^k$ such that*

$$g_c(\omega) = \frac{g(c\omega)}{c}$$

*is such that $\lim_{c \to \infty} \|g_c(\omega) - g_\infty(\omega)\| = 0$ uniformly on compacts for some $h_\infty : \mathbb{R}^k \to \mathbb{R}^k$ and that the origin is the unique and globally asymptotically stable equilibrium of*

$$\dot{\omega} = g_\infty(\omega).$$

*Proof.* Let us expand the definition and obtain

$$\begin{aligned}
g_c(\omega) &= \frac{\mathbb{E}\left[\phi(x,a) \left(\tau^{n,m} - q_{c\omega}(x,a)\right)\right]}{c} \\
&= \frac{\mathbb{E}\left[\phi(x,a)\tau^{n,m}\right]}{c} - \mathbb{E}\left[\phi(x,a)q_\omega(x,a)\right].
\end{aligned}$$

As $c \to \infty$, we have that $g_c(\omega) \to g_\infty(\omega)$ uniformly, using $g_\infty(\omega) = -\mathbb{E}\left[\phi(x,a)q_\omega(x,a)\right]$.

We then have that the ordinary differential equation

$$\dot{\omega} = g_\infty(\omega)$$
$$= -\mathbb{E}\left[\phi(x,a)q_\omega(x,a)\right]$$
$$= -\mathbb{E}\left[\phi(x,a)\phi^T(x,a)\right]\omega$$

is linear and time-invariant. Since the matrix above is invertible, we have that the origin is its unique and globally asymptotically stable equilibrium. We conclude the result. $\qquad\square$

## C.2 Performance

We establish the following lemma, quantifying the bias between the multi $Q$-learning target and the target from the multi-Bellman operator. We will use the lemma in the subsequent theorem, which will include function approximation.

**Lemma C.7.** *We have that*

$$0 \leq \mathbb{E}\left[\tau^{n,m}(x,a) - \left(\mathrm{H}^n q\right)(x,a)\right] \leq \sqrt{|\mathcal{A}|-1}\sum_{i=1}^{n}\gamma^{n-i}\frac{\vartheta^i}{\sqrt{m}},$$

*where $\vartheta^n = \max\sqrt{\mathrm{var}\left(\tau^{n-1,m}\right)}$ and $\vartheta^1 = 0$*

*Proof.* We prove by induction. The basis is true because the bias of the 1-step update is zero. We move on to the step.

From the definition and then taking algebraic operations, we have,

$$\mathbb{E}\left[\tau^{n+1,m}(x,a) - \left(\mathrm{H}^{n+1}q\right)(x,a)\right] = \mathbb{E}\Bigg[\left(r(x,a) + \gamma\max_{a'}\left(\frac{1}{m}\sum_{i=1}^{m}\tau^{n,m}\left(x',a'\right)\right)\right) -$$
$$- \left(r(x,a) + \gamma\max_{a'}\left(\left(\mathrm{H}^n q\right)\left(x',a'\right)\right)\right)\Bigg]$$
$$= \gamma\mathbb{E}\left[\max_{a'}\frac{1}{m}\sum_{i=1}^{m}\tau^{n,m}\left(x',a'\right) - \max_{a'}\left(\left(\mathrm{H}^n q\right)\left(x',a'\right)\right)\right]$$

Now, we use a result from Aven (1985) that says that, for given set of random variables $\{\zeta_i\}_{i \leq I}$

$$\mathbb{E}\left[\max_i \zeta_i\right] \leq \max_i \mathbb{E}\left[\zeta_i\right] + \sqrt{\frac{I-1}{I}\sum_i \mathrm{var}\left(\zeta_i\right)},$$

and have, after Jensen's inequality exchanges the order of the max and the expectation, that

$$\mathbb{E}\left[\tau^{n+1,m}(x,a) - \left(\mathrm{H}^{n+1}q\right)(x,a)\right] \leq \gamma\mathbb{E}\left[\max_{a'}\mathbb{E}\left[\tau^{n,m}\left(x',a'\right)\right] - \max_{a'}\left(\mathrm{H}^n q\right)\left(x',a'\right)\right]$$
$$+ \gamma\sqrt{\frac{|\mathcal{A}|-1}{|\mathcal{A}|}\sum_{a'}\mathrm{var}\left(\frac{1}{m}\sum_{i=1}^{m}\tau^{n,m}\left(x',a'\right)\right)}$$

Then, we add the absolute value and see that

$$\left|\mathbb{E}\left[\tau^{n+1,m}(x,a) - \left(\mathrm{H}^{n+1}q\right)(x,a)\right]\right| \leq \gamma\mathbb{E}\left[\left|\max_{a'}\mathbb{E}\left[\tau^{n,m}\left(x',a'\right)\right] - \max_{a'}\left(\mathrm{H}^n q\right)\left(x',a'\right)\right|\right]$$
$$+ \gamma\sqrt{\frac{|\mathcal{A}|-1}{|\mathcal{A}|}\sum_{a'}\mathrm{var}\left(\frac{1}{m}\sum_{i=1}^{m}\tau^{n,m}\left(x',a'\right)\right)}$$

$$\leq \gamma \mathbb{E}\left[\max_{a'} \mathbb{E}\left[\tau^{n,m}\left(x',a'\right) - \left(\mathrm{H}^n q\right)\left(x',a'\right)\right]\right]$$

$$+ \gamma \sqrt{\frac{|\mathcal{A}|-1}{|\mathcal{A}|} \sum_{a'} \mathrm{var}\left(\frac{1}{m}\sum_{i=1}^m \tau^{n,m}\left(x',a'\right)\right)}$$

$$\leq \gamma \mathbb{E}\left[\max_{a'} \mathbb{E}\left[\tau^{n,m}\left(x',a'\right) - \left(\mathrm{H}^n q\right)\left(x',a'\right)\right]\right]$$

$$+ \gamma \sqrt{\frac{|\mathcal{A}|-1}{|\mathcal{A}|}|\mathcal{A}| \max \mathrm{var}\left(\frac{1}{m}\sum_{i=1}^m \tau^{n,m}\left(x',a'\right)\right)}$$

Now, we use that for a given random variable $\zeta$,

$$\mathrm{var}\left(\frac{1}{m}\sum_{i=1}^m \zeta\right) = \frac{1}{m}\mathrm{var}\left(\zeta\right)$$

and proceed to see that

$$|\mathbb{E}\left[\tau^{n+1,m}(x,a) - \left(\mathrm{H}^{n+1}q\right)(x,a)\right]| \leq \gamma \mathbb{E}\left[\tau^{n,m}(x',a') - \left(\mathrm{H}^n q\right)\left(x',a'\right)\right]$$

$$+ \gamma \sqrt{\frac{\left(|\mathcal{A}|-1\right)\max \mathrm{var}\left(\tau^{n,m}\left(x',a'\right)\right)}{m}}$$

Next, we use that

$$\sqrt{\max(\cdot)} = \max \sqrt{\cdot}$$

and the hypothesis to conclude

$$|\mathbb{E}\left[\tau^{n+1,m}(x,a) - \left(\mathrm{H}^{n+1}q\right)(x,a)\right]| \leq \gamma\sqrt{|\mathcal{A}|-1}\sum_{i=1}^n \gamma^{n-i}\frac{\vartheta^i}{\sqrt{m}} + \gamma\sqrt{|\mathcal{A}|-1}\frac{\vartheta^{n+1}}{\sqrt{m}}$$

$$\leq \sqrt{|\mathcal{A}|-1}\sum_{i=1}^{n+1}\gamma^{n+1-i}\frac{\vartheta^i}{\sqrt{m}}.$$

$\square$

**Theorem C.8.** *For all $n \geq N$ and $m \in \mathbb{N}$, we have that*

$$\left\|q_{\tilde{\omega}^n} - q_{\hat{\omega}^n,m}\right\| \leq \frac{\xi}{1-\lambda\gamma^n}\frac{\vartheta^n}{\sqrt{m}},$$

$\vartheta^n = \max \sqrt{\mathrm{var}\left(\tau^{n-1,m}\right)}$, $\xi = \phi_{\max}^2 \sigma_{\max}\sqrt{|\mathcal{A}|-1}$.

*Proof.*

$$\left\|q_{\tilde{\omega}^n} - q_{\hat{\omega}^n,m}\right\|_\infty \leq \left\|\mathrm{Proj}\,\mathrm{H}^n q_{\tilde{\omega}^n} - \mathrm{Proj}\,\mathrm{H}^n q_{\hat{\omega}^n,m}\right\|_\infty + \left\|\mathrm{Proj}\,\mathrm{H}^n q_{\hat{\omega}^n,m} - q_{\hat{\omega}^n,m}\right\|_\infty$$

$$\leq \lambda\gamma^n\left\|q_{\tilde{\omega}^n} - q_{\hat{\omega}^n,m}\right\|_\infty + \left\|\mathrm{Proj}\,\mathrm{H}^n q_{\hat{\omega}^n,m} - q_{\hat{\omega}^n,m}\right\|_\infty.$$

$$\left\|\mathrm{Proj}\,\mathrm{H}^n q_{\hat{\omega}^n,m} - q_{\hat{\omega}^n,m}\right\|_\infty \leq \phi_{\max}^2 \sigma_{\max}|\mathbb{E}\left[\left(\mathrm{H}^n q_{\hat{\omega}^n,m}\right)(x,a) - \left(\tau^n q_{\hat{\omega}^n,m}\right)(x,a)\right]|$$

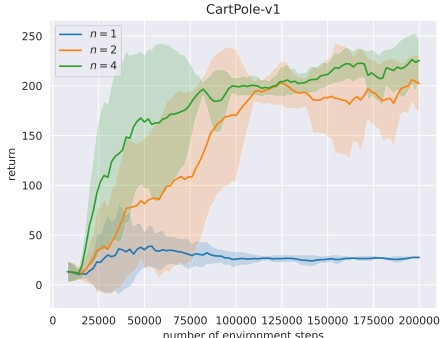

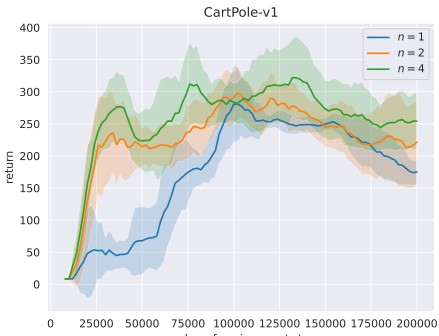

(a) Cartpole, with each dimension of the state space discretized in two.

(b) Cartpole, with each dimension of the state space discretized in four.

Figure 7: Cartpole results. The $y$-axis shows the average return. The performance increases with $n$.

Using the previous lemma,

$$\left| \mathbb{E}\left[ \left( \mathrm{H}^n q_{\hat{\omega}^{n,m}} \right)(x,a) - \left( \tau^n q_{\hat{\omega}^{n,m}} \right)(x,a) \right] \right| = \mathbb{E}\left[ \left( \tau^n q_{\hat{\omega}^{n,m}} \right)(x,a) - \left( \mathrm{H}^n q_{\hat{\omega}^{n,m}} \right)(x,a) \right]$$

$$\leq \sqrt{(|\mathcal{A}|-1)} \sum_{i=1}^{n} \gamma^{n-i} \frac{\vartheta^i}{\sqrt{m}}$$

$$\leq \sqrt{(|\mathcal{A}|-1)} \frac{\vartheta^n}{\sqrt{m}} \sum_{i=1}^{n} \gamma^{n-i}$$

$$\leq \frac{\sqrt{(|\mathcal{A}|-1)}}{1-\gamma} \frac{\vartheta^n}{\sqrt{m}}$$

where we used that, for all $i \leq n$

$$\vartheta^i \leq \vartheta^n$$

and

$$\sum_{i=1}^{n} \gamma^{n-i} = \frac{1-\gamma^{n-1}}{1-\gamma} \leq \frac{1}{1-\gamma}.$$

We conclude the result. □

## D Additional experiments

In this appendix we provide additional results regarding the Carpole environment. We show results for features resulting from a finer discretization of the state space in Figure 7. Specifically, each dimension of the state space is discretized in four, whereas in the results in the main text each dimension is discretized in two. We recall the results from the main text for comparison. We see that the performance of $Q$-learning ($n = 1$) improved the most. The performance of multi $Q$-learning ($n = 2$ and $n = 4$) also improves. Overall the performance of multi $Q$-learning is superior to $Q$-learning.

