# OpenReview forum: "Multi-Bellman operator for convergence of $Q$-learning with linear function approximation"
_TMLR — Accepted by TMLR_

### Review · Reviewer_j3Jj · 2024-12-01

**Summary Of Contributions:**

This paper studies the problem of convergence of Q-learning with linear function approximation. The authors suggest that using a so-called multi-Bellman operator method resolve the divergence issue. The key idea is that if we use a $n$-step roll of Bellman operator, the contraction factor can be controlled by $\gamma^n$ where $n$ is the number of steps and $\gamma$ is the discount factor. A sufficiently large $n$ will ensure the contraction property of the suggested operator.


Furthermore, the convergence of the proposed stochastic algorithm is established from the ODE argument by Borkar-Meyn Theorem. The proposed stochastic algorithm uses a biased estimate of multi Bellman operator $H^nq_{w}$, which requires knowledge of the reward function.

Lastly, the authors study the bias of the solution of multi-Bellman equation (Theorem 3.7); bias of the multi Q-learning algorithm which is due to using a biased estimator of multi-Bellman operator.


Moreover, the experimental results are provided.

**Audience:**

Yes

**Claims And Evidence:**

Yes

**Requested Changes:**

1. Can the authors comment more about the bound in Theorem 3.6? For example, when $n\to 1$, do we recover the function approximation error $\frac{1}{1-\gamma}||q^*-\mathrm{Proj}(q^*)||$?

2. As for the Cartpole environment, Can the authors also provide comparison under the case when Q-learning performs well?

3. It would be helpful to understand the algorithm if the authors can add a pseudo-code.

4. I think there is typo in the proof of Lemma C.7. In the second line of the first equation: $Hq$ should be $H^{n}q$.

5. There is slight mistake in proof of Lemma C.5. $\phi^2 \gamma^n-\lambda_{\max}$ should be $\phi^2 \gamma^n-\lambda_{\min}$.

**Strengths And Weaknesses:**

**Strength:**

1. The authors tackle an important problem, the divergence of Q-learning under linear function approximation, and using a multi-step method provides a new perspective on this problem.

2. The theoretical analysis is solid.


**Weakness:**


1. The rollout of $n$-step method has been well-studied (as noted in the related works) and its contraction rate being $\gamma^n$ has been well-known in the literature as noted in the related works.

2. The practical implementation is somewhat difficult. The algorithm converges to a biased solution, and to reduce this bias, we need a large number of $n$-step sequence each state pair. Furthermore, we need the knowledge of reward function because we need to take a maximum over reward function to compute $\tau^{n,m}$.

---

### Review · Reviewer_o6SM · 2024-12-21

**Summary Of Contributions:**

The paper proposes a multi-step bellman operator based $Q$-learning algorithm with linear function approximation. Instead of applying 1-step greedy $Q$-value updates using Bellman Equation, this paper proposes to updates the $Q$-value function iteratively $n$ times over every step. Further, the paper proves that the proposed method allows for projection of the Bellman operator to contract for sufficiently large values of $n$. Additionally, the paper implements the algorithm on standard RL benchmarking environments (acrobat and cartpole) to empirically show the performance of the algorithm.

**Audience:**

Yes

**Claims And Evidence:**

Yes

**Requested Changes:**

1. Please conduct an the empirical study by updating the policy after every $m$ steps to update $Q$-values using $m$ iid data points. It will be a impactful if the algorithm convergence can be improved by reducing the variance of multi-q learning.

2. Please correct the update equations for $\tau_t^{n+1, m}$

3. Please add the proposed algorithm explicitly.

4. Please use equation number so that the equations can be easily referred.

**Strengths And Weaknesses:**

Strengths:
1. Proposal of updating Q-values multiple times in a single step is novel and interesting.
2. The proposed use of multi-Bellman operator and its application to obtain a contraction of the Projection of Bellman operator is interesting.


Weaknesses:
1. For learning Q-value function given the complete MDP, Bellman equation is applied multiple times which essentially is the multi-Bellman operator. Essentially, the proposed algorithm solves for Q-value function with unknown dynamics as if were completely known. This needs to be discussed in the paper if and how the multi-Bellman operator is different from this.

2. Multi step $Q$-learning equation (above section 4.1) is not clear. Which variable contains $i$ in the update equation for $\tau_t^{n+1,m}$.

3. This statement, "At time step $t$, instead of building a 1-step greedy target, multi Q-learning builds a target that is obtained
by trying, on every state encountered along $n$-step trajectory starting at $x'_t$, every action $m$ times, and maximizing the average of the sum of the discounted rewards at each level.". How does this not result an $m^n$ exponential complexity. This is not clear.

4. "We assume access to a fixed replay buffer of 1-step transitions and a simulator to sample transitions" If you have access to simulator, why the simulator alone cannot be used for learning is not clear?

---

### Review · Reviewer_ScuR · 2025-01-29

**Summary Of Contributions:**

The paper addresses the convergence issues of Q-learning with linear function approximation. The authors introduce a multi-Bellman operator and the multi-Q-learning algorithm. The algorithm is novel and theoretically well-motivated. Finally, the authors show the effectiveness of their algorithm empirically.

**Audience:**

Yes

**Claims And Evidence:**

Yes

**Requested Changes:**

* Move Figure 3 to the top of the page to make the paper easier to read.
* Add a graphical representation of the toy environments.
* Motivate the "Bias" example. Why is this example necessary?
* Move the "Bias" example into 5.1 since it is about convergence and not performance.
* Equation at the top of page 5. "i" is not used inside of the sum.
* Why are so many discount factors are used in the experiments? I see that 0.9, 0.98 and 0.99 are used without justification.
* Mention semi-gradient methods in the background, since they are mentioned in the related work.

**Strengths And Weaknesses:**

## Strengths
* The experiments, while limited, are convincing.
* The limitations are explicitly discussed in 7.1.
* The multi-Q-learning algorithm is novel and theoretically supported.
* Convergence of Q learning is of interest to the machine learning community.

## Weaknesses
* Very limited discussion of the "Bias" example.
* The paper lacks context around examples. Why are these examples used?
* The usefulness of this algorithm in real world scenario is limited. (but correctly highlighted in the limitations)

---

### Decision · Action_Editor_daGP · 2025-03-18

**Recommendation:** Accept as is

**Comment:**

All reviewers unanimously agree to accept the paper. They find the perspective of viewing the divergence of Q-learning from n-step method to be new and expect that this will be helpful to the community. The theoretical results including bias and variance analysis is also appreciated and is expected to be helpful to understand the limits of the n-step method of Q-learning.

A small comment is regarding the focus on linear function approximation and the concern that it may not align with ICLR's emphasis on deep (reinforcement) learning algorithms. I do not believe this to be a big issue. Another concern is regarding the exponential complexity of the algorithm, which authors may want to comment on in the camera-ready version (perhaps in the discussion). A reviewer also points that this work might be useful in training LLMs using a reward setup at each token which can act as a simulator.

**Audience:**

Yes, the RL community will benefit from this work.

**Claims And Evidence:**

The claim is that using multi-Bellman operator can have a fixed point even when the single-step Bellman may not. This is proved under some assumption. Reviewers find the claim to be correct and experimental evidence to be enough.